# Degradation Mechanism of Autophagy-Related Proteins and Research Progress

**DOI:** 10.3390/ijms23137301

**Published:** 2022-06-30

**Authors:** Yanhui Zhou, Hakim Manghwar, Weiming Hu, Fen Liu

**Affiliations:** 1Lushan Botanical Garden, Chinese Academy of Sciences, Jiujiang 332900, China; zhouyanhui994@163.com (Y.Z.); hakim@lsbg.cn (H.M.); 2School of Life Sciences, Nanchang University, Nanchang 330031, China

**Keywords:** autophagy-related protein, degradation, ubiquitin, proteasome, autophagy

## Abstract

In all eukaryotes, autophagy is the main pathway for nutrient recycling, which encapsulates parts of the cytoplasm and organelles in double-membrane vesicles, and then fuses with lysosomes/vacuoles to degrade them. Autophagy is a highly dynamic and relatively complex process influenced by multiple factors. Under normal growth conditions, it is maintained at basal levels. However, when plants are subjected to biotic and abiotic stresses, such as pathogens, drought, waterlogging, nutrient deficiencies, etc., autophagy is activated to help cells to survive under stress conditions. At present, the regulation of autophagy is mainly reflected in hormones, second messengers, post-transcriptional regulation, and protein post-translational modification. In recent years, the degradation mechanism of autophagy-related proteins has attracted much attention. In this review, we have summarized how autophagy-related proteins are degraded in yeast, animals, and plants, which will help us to have a more comprehensive and systematic understanding of the regulation mechanisms of autophagy. Moreover, research progress on the degradation of autophagy-related proteins in plants has been discussed.

## 1. Introduction

Autophagy widely exists in eukaryotic cells, which is a relatively conservative process in evolution. It is responsible for the transport of certain cytoplasmic proteins and subcellular organelles into lysosomes/vacuoles for degradation, thereby contributing to the recycling of intracellular nutrients [1,2]. In yeast and mammals, autophagy consists of three types: microautophagy, macroautophagy, and chaperone-mediated autophagy (CMA). However, CMA is absent in plants and has been replaced by mega-autophagy [3]. Among them, the so-called autophagy usually refers to macroautophagy, which contains both selective and non-selective intracellular degradation processes [4] (Figure 1). Selective autophagy is distinguished from bulk autophagy by the use of selective autophagy receptors (SARs) [5]. Macroautophagy begins with a series of autophagy-related (ATG) proteins forming phagophore structures at the assembly site of the phagophore, and then recruiting substrate molecules, and forming autophagosome with double-layer membrane structure through the expansion and closure of vesicles. Subsequently, the outer membrane of the autophagosome fuses with the lysosomal or tonoplast membrane to release the contents into the lysosomal lumen or vacuole in the form of an autophagic body with only a single membrane, and is degraded into small molecular substances for recycling under the role of acid hydrolase [6,7,8]. Microautophagy is a process that isolates and uptakes cell components by direct enclosure with the lysosomal/vacuolar membrane [9]. Autophagy plays an important role in maintaining cellular homeostasis. Plants are exposed to many biotic and abiotic stresses, and these stresses affect plants [10,11,12,13,14,15,16]. When cells face stress conditions, such as nutrient deprivation, oxygen deficiency, and endoplasmic reticulum (ER) damage, autophagy is highly induced to maintain metabolic and energy balance [16,17,18,19]. During starvation, lysosomal activity is also improved to support autophagic flux [20].

Autophagy is a complex pathway that is tightly regulated. The expression of autophagy-related genes has spatiotemporal specificity and is a dynamic process. Regulation of autophagy-related proteins at different levels, including transcription, post-transcription, translation, and post-translation, which helps to inhibit or activate autophagy [21,22,23]. At the transcriptional level, when plants face drought stress, overexpression of heat shock factor A1a (HsfA1a) can increase the mRNA levels of ATG10 and ATG18f, and the number of autophagosomes is also increased accordingly [24]. In mammals, microRNAs (miRNAs) regulate autophagy at the post-transcriptional level [25]. Post-translational modifications, such as phosphorylation, lipidation, and ubiquitination of autophagy-related proteins, are critical in regulating autophagic activity and duration [23]. For example, AMP-activated protein kinase (AMPK) phosphorylates BECN1 (beclin 1, vacuolar protein sorting (Vps) 30/Atg6 in yeast, and ATG6 in plants) at the site of Thr388 to induce autophagy [26]. ATG4a is modified by hydrogen sulfide (H2S) persulfidation, inhibiting protease activity, thereby negatively regulating autophagy [27]. During the ubiquitination modification process, autophagy-related proteins are labeled with ubiquitin, which is recognized and degraded by the 26S proteasome [28], ultimately inhibiting the production of autophagic vacuoles. In recent years, the regulatory pathways of autophagy-related protein degradation have attracted much attention. In this review, we have mainly focused on the recent advances in core autophagy-related protein degradation mechanisms.

## 2. ATG Complexes in Plants and Animals

ATG proteins play an indispensable role in the process of autophagy in plant and animal cells. According to their diverse functions in different stages of autophagy, core ATG proteins can be divided into four major complexes (Figure 2).

### 2.1. ATG1/ATG13 Protein Kinase Complex

The ATG1/ATG13 protein kinase complex is located at the most upstream position in the ATG proteins recruitment hierarchy and initiates the formation of phagophores in response to the demand for nutrients [29]. When cells are under normal growth conditions, the target of rapamycin (TOR) kinase complex hyperphosphorylates the ATG13, reducing its affinity for ATG1 (Unc-51-like autophagy activating kinases 1, ULK1 in mammals), thereby inhibiting autophagy. When plants are placed under stress conditions, TOR kinase inactivation dephosphorylates the ATG13, promoting its tight binding to ATG1. Then, ATG1 kinase activates to autophosphorylate itself, resulting in ATG1, ATG11, ATG13, and ATG101 forming a complex and connecting to the phagophore membrane, which promotes several downstream autophagy steps [30]. In mammalian cells, the ULK complex is made up of ULK1/2, ATG13, ATG101, and FIP200/RB1CC1 (Atg11 and Atg17 in yeast) [8]. FIP200/RB1CC1 is the scaffold protein of ULK1/2 and ATG13, which is essential for the phosphorylation and stability of ULK1/2 [31]. In addition to TOR complexes, several pathways, including cAMP-dependent protein kinase A and AMP-activated protein kinase, can also regulate the ATG1/ULK complex [32].

### 2.2. ATG9/2/18 Transmembrane Complex

The ATG9/2/18 transmembrane complex participates in the formation of phagophore and membrane fusion. ATG9 (ATG9A in mammals) is the only transmembrane protein in the autophagy process of eukaryocytes, which provides a membrane source for forming pre-autophagosomal structure (PAS) and driving the extension of the isolation membrane. It has been found that ATG9 loses contact with the phagophore membrane in the process of autophagy, thus regulating the formation of phagophores from plant ER [33]. ATG2 (ATG2A in mammals) interacts with ATG18 (WD-repeat protein interacting with phosphoinositides, WIPI in mammals), both of which act in the later stages of phagosome formation, fixing the PAS/isolation membrane on the ER and transferring lipids, effectively completing the closing process [34].

### 2.3. Phosphatidylinositol-3-Kinase (PI3K) Complex

The PI3K complex mediates vesicle nucleation. The PI3K complex can be further divided into complexes I and II. In plants, complex I consists of ATG6, ATG14 (ATG14L in mammals), VPS15, and VPS34, and complex II consists of ATG6, VPS15, VPS34, and VPS38. The PI3K complex and phosphatidylinositol-3-phosphate (PI3P) are responsible for the modification of de novo synthesis of the phagophore, and PI3P recruits the ATG2-ATG18 complex to the phagophore membrane and participates in the extension of the phagophore [35].

### 2.4. ATG5/ATG12 and ATG8-Phosphatidylethanolamine (PE) Conjugation Systems

The ATG5-ATG12 and ATG8-PE ubiquitin-like conjugating systems not only regulate the initiation of phagophore formation but also function at a downstream step [36]. Among them, ATG4 is a key Cys-protease with important functions for ATG8 (the light chain 3, LC3 in mammals) lipidation and free ATG8 turnover [37]. During the process of autophagy, the C-terminus of ATG8 is first recognized and cleaved by ATG4 protease. Subsequently, ATG7 with ubiquitin-activating enzyme E1 activity binds to ATG8 and ATG12, activating their mature forms. The activated ATG8 and ATG12 transfer to ATG3 and ATG10 with ubiquitin-binding enzyme E2 activity, respectively, and finally link to the substrate. Among them, ubiquitination folding protein ATG8 is combined with PE, and ATG12 is coupled with ATG5 to form a complex. In addition, the ATG12-ATG5 adduct combines with ATG16 (ATG16L1 in mammals) to form oligomeric complexes, which promotes the formation of ATG8-PE adducts [38].

## 3. Protein Degradation Pathways in Eukaryotic Cells

To ensure that the autophagy machinery can respond in time to each stimulus, regulating the rapid inactivation/degradation of autophagy-related proteins through negative feedback would be an ideal mechanism. In eukaryotes, autophagy and the ubiquitin-proteasome system (UPS) maintain proteome homeostasis by degrading redundant or misfolded proteins [39]. Among them, autophagy mainly degrades long-lived proteins and damaged or redundant organelles, and the UPS mainly impairs short-lived proteins [40,41]. Ubiquitination of proteins involves multiple steps. First, ubiquitin-activating enzyme E1 activates the C-terminus of ubiquitin molecules by consuming adenosine triphosphate (ATP) to form an acyl-adenylate complex. Second, the ubiquitin-conjugating enzyme E2 transduces activated ubiquitin molecules through cysteine residues. The ubiquitin ligase E3 then links the ubiquitin molecule to the amino group of the lysine side chain of the target protein by catalyzing the formation of isopeptide bonds. The above steps are repeated to form ubiquitinated target proteins with oligomeric ubiquitin chains in order to form ubiquitination tags. Finally, the proteasome cap recognizes the ubiquitination tag and uses ATP hydrolysis to provide energy to drive ubiquitin molecular excision and target protein unfolding. Then, the unfolded target protein is transferred to the proteasome core for degradation [42].

Ubiquitination-modified proteins can be deubiquitinated by the family of deubiquitinases (DUBs), which hydrolyze the isopeptide bonds formed by target protein lysine residues and ubiquitin molecules, thus inversely regulating the degradation of protein [43]. Thus far, six structurally distinct DUB families have been described, namely, the ubiquitin-specific proteases (USPs), the Josephin family, the ubiquitin C-terminal hydrolases (UCHs), ovarian tumor-related proteases (OTUs), a family of Zn-dependent JAB1/MPN/MOV34 metalloprotease DUBs (JAMMs), and the motif interacting with ubiquitin (MIU)-containing novel DUB family (MINDYs) [44]. Therefore, ubiquitination is a reversible and controllable process. Autophagy and the UPS work together to control intracellular protein quality at the stage of individual development. Plant autophagy-related proteins can be degraded by the UPS pathway [45], while components of the UPS, such as the 26S proteasome, can also be degraded by autophagy [46].

## 4. Degradation of Yeast Autophagy-Related Proteins

Among yeast autophagy-related proteins, the degradation mechanisms of the transmembrane proteins Atg9 and Atg32 have been discovered. The content of Atg9 in cells is known to be a key factor in determining the number of phagosomes [47]. Under normal growth conditions, Atg9 is ubiquitinated at the Lys113, Lys121, and Lys138 sites and subsequently degraded by the proteasome at peripheral sites away from PAS by Met30, which is one of the substrate recognition components of the ubiquitin ligase SCF complex (Skp, Cullin, F-box containing) (Table 1) [48,49]. When cells are under nutrient-deficient conditions, Atg9 is stabilized by Atg1 phosphorylation, thus inhibiting the degradation process, and activating the autophagy process [50].

In yeast mitophagy, the mitochondrial outer membrane protein Atg32 is the only specific receptor [51], which determines mitochondrial turnover. Overexpression of Atg32 can effectively activate mitophagy, while the deletion of Atg32 leads to a defect in mitophagy [52,53]. Interestingly, two published papers have different findings on the degradation mechanism of Atg32. Levchenko et al. [54] reported that there are two forms of Atg32 in *Saccharomyces cerevisiae*, namely, unmodified Atg32 and post-translation modified Atg32, and their degradation pathways are different. Post-translationally modified Atg32 is activated when mitophagy is induced to target mitochondria for degradation. Experiments show that this modification is neither phosphorylation nor ubiquitination. In contrast, the turnover of unmodified Atg32 is mediated by an unknown protease independent of proteasome and vacuolar proteases [54]. However, studies by Camougrand et al. [55] demonstrated that the degradation of Atg32 is associated with the UPS. Ubiquitination of Atg32 occurs at least on Lys282 residues and is mediated by several E3 ligases, including Rsp5, which is a relatively complicated process [55]. To sum up, apart from the different strains used in the experimental materials, these two contradictory observations may be attributed to the differences in growth conditions or mitophagy induction, leading to the coordinated regulation of Atg32 levels by multiple pathways. Therefore, further research is needed to determine the details of different degradation pathways of Atg32.

## 5. Degradation of Autophagy-Related Proteins in Metazoa

### 5.1. Degradation of ULK Complex Components

In mammals, the ULK complex includes four parts: ULK1/2, ATG13, RB1CC1/FIP200, and ATG101. Among them, ULK1 plays a major role in autophagy [74]. The level of ULK1 is regulated by E3 ubiquitin ligase and USPs. Under prolonged starvation, autophosphorylation of ULK1 at the Ser1042/Thr1046 site facilitates its recruitment to Kelch-like (KLHL) 20 for ubiquitination and protein degradation as a substrate for the Cullin3 (Cul3)-KLHL20 ubiquitin ligase, thereby preventing cell unrestricted activate autophagy. In addition, KLHL20 also coordinates the degradation of ATG13 through an indirect mechanism [56]. Cancer can hijack the protective function of autophagy to promote tumorigenesis [75]. In breast cancer cells, mitogen-activated protein kinase (MAPK) 1/3 kinases promote its binding to the E3 ligase BTRC by phosphorylating ULK1 at multiple sites, triggering proteasomal degradation of K48-linked ULK1 ubiquitination, ultimately attenuating mitophagy and promoting breast cancer bone metastases [57]. Tumor necrosis factor receptor-associated factor 3 (TRAF3) is a member of the TRAF family with E3 ligase activity. In macrophages, TRAF3 mediates K48-linked ULK1 ubiquitination and proteasomal degradation, and the TRK-fused gene (TFG)-TRAF3 complex is able to interfere with TRAF3-ULK1 interaction to stabilize ULK1 in response to lipopolysaccharide (LPS)-induced pyroptosis [58]. Neural precursor cell expressed developmentally downregulated 4-like (NEDD4L) is an E3 ubiquitin ligase containing a HECT domain. When cells need to endure starvation, NEDD4L ubiquitinates ULK1 at the Lys925 and Lys933 sites. It induces its degradation by the proteasome, thus autophagy is calibrated to optimal levels to ensure the survival of cells. Interestingly, NEDD4L-mediated ubiquitination of ULK1 is not a canonical K48 linkage, but a K27 and K29 linkage [59].

The degradation of ULK1 was also regulated by deubiquitination. USP20 mediates ULK1 deubiquitination in the basal state, thus interfering with lysosome-dependent degradation of ULK1 and playing a key role in autophagy initiation [76]. ATG101 is a scaffold protein that maintains the stability of ATG13 within the ULK1/ATG1 complex [77,78]. The HECT, UBA, and WWE domain containing E3 ubiquitin-protein ligase 1 (HUWE1) regulates cell proliferation and cell death and is an important factor affecting tumorigenesis [79]. In cancer cells, HUWE1 targets the ubiquitination and degradation of the C-terminal region of ATG101, inhibiting autophagy activity to reduce cancer cell survival [60].

### 5.2. Degradation of ATG9 Complex Components

The mammalian ATG9 complex contains four ATG18 homologs, namely, WIPI 1-4. Studies have shown that WIPI2 can associate the phagophore with ER [80] and play a role in cellular antibacterial autophagy [81]. Under basal cellular conditions, mTORC1 mediates the phosphorylation of WIPI2 at the Ser395 site, thereby enhancing the specific interaction of WIPI2 with HUWE1, promoting the ubiquitination and proteasomal degradation of WIPI2, and inhibiting the degree of autophagy [61].

### 5.3. Degradation of PI3K Complex Components

In mammals, the autophagy-specific PI3K complex I consists of four proteins: VPS34, VPS15, BECN1, and ATG14L [82]. Ubiquitination and proteasomal degradation of ATG14L are regulated by the E3 ubiquitin ligase complex zinc finger and BTB domain containing 16 (ZBTB16)-CUL3-Regulator of cullins 1 (Roc1). Under normal nutritional conditions, inhibition of G-protein-coupled receptors (GPCRs) activates glycogen synthase kinase-3β (GSK-3β). It mediates ZBTB16 phosphorylation to promote its autoubiquitination, which in turn upregulates ATG14L levels to activate autophagy [62]. In host cells, *Streptococcus pneumoniae* (Sp) releases choline binding protein C (CbpC) to form the ATG14L-CbpC-sequestosome 1 (SQSTM1)/p62 intracellular complex, promoting autophagy-dependent degradation of ATG14L, which results in inhibiting autophagosome-lysosome fusion and bactericidal autophagic degradation, increasing bacterial survival [83].

BECN1, the first tumor-associated ATG protein discovered in mammals, plays a role in the formation, elongation, and maturation of autophagosomes [84,85]. Different E3 ligases play various roles in regulating BECN1 activity by binding to specific types of ubiquitin chains. NEDD4 and Ring finger protein 216 (RNF216) modify BECN1 through K11, K63 and K48-linked polyubiquitination chains, respectively, which mediates the BECN1 degradation in a proteasome-dependent manner and negatively regulates autophagy [63,64]. Similarly, CUL3-KLHL20-mediated ubiquitination of BECN1 is primarily responsible for the termination of autophagy during prolonged starvation, whereas CUL3-KLHL38-mediated BECN1 K48-linked ubiquitination prevents autophagy under normal conditions [65]. TRAF6 acts as an E3 ligase to trigger the K63-linked ubiquitination of BECN1, while the deubiquitinating enzyme A20 inhibits the ubiquitination of BECN1 [66]. Furthermore, tripartite motif 59 (TRIM59) mediates K48-linked ubiquitination and proteasomal degradation of TRAF6, which results in affecting TRAF6 to ubiquitinate BECN1, regulating the initiation of autophagy [86]. In contrast, three ubiquitin-specific peptidases, USP10, USP13, and USP19, act as positive regulators of autophagy and maintain BECN1 by mediating deubiquitination of BECN1 in the VPS34 complex stability [87,88]. By binding to BECN1, solute carrier family 9 isoform 3 regulator 1 (SLC9A3R1) inhibits BECN1 ubiquitination and proteasomal degradation, which stimulates the formation of the autophagic core lipid kinase complex [89].

VPS34 is the only mammalian phosphatidylinositol 3-kinase class III (PI3KC3) that is activated on phagophore and endosomal membranes [90,91]. The level of VPS34 in cells is controlled by a variety of regulatory factors. For example, the ubiquitin-protein ligase E3C (UBE3C) and the deubiquitinating enzyme TRABID act antagonistically in regulating the levels of VPS34 to balance autophagic activity. UBE3C assembles K29/K48-branched ubiquitin chains on VPS34, enhancing VPS34 binding to the proteasome for degradation. TRABID stabilizes VPS34 by reducing the ubiquitination of VPS34 K29/K48, which in turn promotes phagophore formation [67]. Similarly, auto-ubiquitination of NEDD4-1 enables it to act as a scaffolding protein to recruit USP13 to form the NEDD4-1-USP13 deubiquitination complex. This complex then promotes autophagy by removing K48-linked polyubiquitin chains on VPS34 [92]. In *Caenorhabditis elegans*, the K63-linked polyubiquitination of VPS34 mediated by the E2 enzyme UBC-13/UEV-1 and the E3 enzyme CHN-1 can stabilize the level of VPS34, promoting phagophore maturation [93]. In addition, phosphorylation of substrates can enhance their recognition by SCF complex [94]. Phosphorylation of VPS34 at the Thr159 site promotes F-Box and leucine-rich repeat protein 20 (FBXL20) as an adaptor for the SCF complex, regulating the ubiquitination and proteasomal degradation of VPS34 [68]. Notably, Tang et al. highlighted that the ubiquitination modification of VPS34 is not directly linked to its degradation process [95].

### 5.4. Degradation of the ATG12-Conjugation System Components

The ATG12-conjugation system includes ATG5, ATG7, ATG10, ATG12, and ATG16L1. ATG12 is a member of the ubiquitin-like protein (UBL) family, which consists of about 20 members [96]. Free ATG12 is highly labile and can be targeted for proteasomal degradation through ubiquitin-dependent or ubiquitin-independent mechanisms [97]. ATG5 is a ubiquitin-like ligase that is conjugated by ATG12 [98]. In Lepidoptera, there is an interaction between ATG1 and ATG5, and *Spodoptera litura* ATG1 (SlATG1) promotes the degradation of ATG5 [99]. In cardiomyocytes, the immunoproteasome β5i subunit interacts with ATG5 to promote the ubiquitination and degradation of ATG5, thereby inhibiting autophagy and leading to cardiac hypertrophy [100]. Moreover, saturated fatty acid palmitate induces ER stress and degrades ATG5 protein through the ER-associated protein degradation (ERAD) pathway, consequently inhibiting autophagy and inducing apoptosis [101].

ATG16L1 interacts with ATG5, which stimulates the ATG8-PE coupling reaction [31]. Gigaxonin (GAN)-E3 ligase ubiquitinates ATG16L1 through K48-type ubiquitin chain polymerization, driving its degradation, thereby controlling the steady-state level of ATG16L1 to ensure fine-tuning of autophagy activation. Loss of GAN leads to the formation of ATG16L1 aggregates that impair phagophore elongation, inhibiting autophagic flux [69].

### 5.5. Degradation of the LC3-Conjugation System Components

In mammalian cells, LC3, GABA type A receptor-associated proteins (GABARAPs), GATE16, and ATG8L exist as yeast Atg8 homologs [102,103]. All of them can be used as autophagosome markers. Under normal conditions, BRUCE acts as an autophagy inhibitor, promoting proteasomal degradation of LC3-I, which results in reducing LC3-II levels and autophagy [104]. Similarly, UBA6 and BIRC6 act synergistically as E1 and E2/E3 enzymes, respectively, for the monoubiquitination and proteasomal degradation of LC3B, which protects cells from cell death caused by excessive autophagy [70,105]. In addition, circumsporozoite protein (CSP) downregulates LC3B through the UPS pathway, but the mechanism involved is unclear [106].

ATG3 acts as an E2-like enzyme in LC3 lipidation. It can autocatalyze itself to form a complex with ATG12 to promote mitochondrial homeostasis [107,108]. Under DNA damage conditions, protein tyrosine kinase 2 (PTK2) phosphorylates ATG3 at the Tyr203 site and promotes the degradation of ATG3 through the ubiquitin-dependent proteasome pathway, resulting in positively regulating the activity of cancer cells [71].

## 6. Degradation of Plant Autophagy-Related Proteins

Up to now, more than 40 kinds of ATG proteins and their related regulatory proteins have been identified in plants [109], mainly from the analysis of yeast autophagy-deficient mutants, but their degradation mechanisms are poorly understood. In *Arabidopsis thaliana*, the turnover of ATG1 and ATG13 is strongly and specifically regulated by nutrition, which closely links autophagy with plant nutritional conditions. During the period of fixed carbon/nitrogen limitation, the levels of ATG1 and ATG13 decreased sharply, but they could be reversed again during refeeding. Studies on *A. thaliana* mutants that damage autophagy or 26S proteasome show that both degradation pathways are involved in their degradation, but autophagy is more directly involved. Under the condition of limited nutrition, ATG1 and ATG13 combine with autophagic-like cytolytic structures and finally transfer to vacuoles together with autophagosomes for degradation [110].

In addition, Liu et al. [35] found a similar phenomenon in their study of ATG14 and its associated PI3K complex. By expressing GFP-ATG14b in *atg7* and *atg14a atg14b* mutants, it was found that there was still free GFP in *atg14a atg14b* mutants, which indicated that ATG14 was degraded, unlike the intact GFP-ATG14b fusion in *atg7* mutants with autophagy defects. Next, after N starvation treatment, co-localization of *atg14a atg14b* mutant roots expressing both mCherry-ATG8a and GFP-ATG14b revealed that ATG14 was bound to autophagic membranes [35]. Thus, similar to ATG1 and ATG13, ATG14 is degraded by the autophagic pathway through associating with the autophagic bodies.

The effect of ubiquitination of autophagy-related protein on autophagy has been widely studied in mammals, but less in plant cells. In plants, a single ubiquitin molecule can form a polyubiquitin chain at a certain site in a target protein (polyubiquitination) or be attached to multiple lysine residues (multi-monoubiquitination). In *A. thaliana*, K48 is commonly used to form polyubiquitin chains [111]. Qi et al. [45,72] successfully studied the turnover of ubiquitination modification of ATG6 and ATG13 in *A. Thaliana* (Figure 3). Under different nutritional conditions, as molecular adaptors, *A. thaliana* TRAF1a and TRAF1b interacted with Ring finger E3 ligase seven in absentia of *Arabidopsis thaliana* 1 (SINAT1)/SINAT2 and SINAT6 to regulate the turnover of ATG6 and ATG13. Among them, SINAT6 contains only a short truncated Ring finger domain compared to SINAT1 and SINAT2 [112]. Under nutrient-rich conditions, *A. thaliana* TRAF1a and TRAF1b interact with SINAT1 and SINAT2 to mediate ubiquitination and degradation of ATG6. However, under conditions of nutrient deprivation, starvation-induced accumulation of SINAT6 reduces the binding of SINAT1 and SINAT2 to ATG6, stabilizes ATG6 levels, and activates autophagy [45].

Similarly, at the Lys607 and Lys609 sites of the ATG13 protein, TRAF1-SINAT1/SINAT2-ATG13 TRAFasome was degraded by ubiquitination modification linked by K48, resulting in the dissociation of ATG1-ATG13 complex to maintain a relatively low autophagy level. However, TRAF1-SINAT6-ATG13 TRAFasome promotes the stability of ATG13, which induces the biogenesis of autophagy in response to nutritional deficiency. Besides, under starvation conditions, ATG1 kinase phosphorylates TRAF1a and promotes its protein stability in vivo, indicating the feedback regulation of autophagy [72]. In short, SINAT1/SINAT2 and SINAT6 play negative and positive roles in regulating the stability of ATG6 and ATG13, thus playing opposite roles in autophagy. Furthermore, at the regulatory level, the turnover of ATG8, which interacts with various adaptor/receptor proteins to recruit specific cargos for degradation, is affected by acyl-coa-binding protein 3 (ACBP3) [113]. In *A. thaliana*, ACBP3, as a phospholipid-binding protein, is involved in the regulation of leaf senescence by regulating membrane phospholipid metabolism and the stability of ATG8. Overexpression of ACBP3 promotes the degradation of ATG8 and disrupts the formation of autophagic vesicles, which results in inhibiting autophagy [114].

Except for maintaining cell homeostasis, autophagy also plays a vital role in plant immunity against pathogens. However, bacteria have evolved the mechanism of evading autophagic clearance in order to better parasitize in the host [115]. Type III effector proteins (T3E) of plant pathogens are present in host cells, and these effectors are capable of manipulating host defense response [116]. Src homology 3 (SH3) domain-containing protein-2 (SH3P2) is a novel membrane-associated protein involved in the formation of autophagosomes [117]. When *Xanthomonas campestris* pv. *vesicatoria (Xcv)* invades plant cells; it utilizes the bacterial effector E3 ligase XopL to mediate the ubiquitination and degradation of SH3P2 in a proteasome-dependent manner, and inhibits host autophagy, thus enhancing the virulence of *Xanthomonas*. Intriguingly, in host cells, XopL is recognized and degraded by NBR1/Joka2-mediated selective autophagy related to defense. Hence, the mutual targeting of pathogen effector XopL and plant protein SH3P2 reveals the complex antagonism between pathogen and plant autophagy mechanism [73].

## 7. Concluding Remarks and Future Perspectives

Autophagy is an important regulatory factor for eukaryotic cells to cope with various stresses. In plants, it is precisely regulated by environmental changes and developmental stages [118]. In this review, the degradation mechanism of autophagy-related proteins in eukaryotes has been discussed. There are many studies on the degradation mechanism of animal autophagy-related proteins, probably because the regulation of autophagy can be used as an effective intervention for the treatment of metabolic and neurodegenerative diseases [119,120]. However, there is little information about how autophagy-related proteins are degraded in plant cells, and only the degradation mechanisms of ATG1, ATG6, ATG13, ATG14, and SH3P2 have been discovered thus far [45,72,110].

As far as the known degradation mechanism of animal autophagy-related proteins is concerned, ubiquitin-proteasome mainly plays a role in it. What is more, some regulatory factors that affect the degradation of autophagy-related proteins were also found. Both autophagy and the UPS are involved in the degradation of the plant ATG1-ATG13 complex. Then, how plants themselves coordinate the effects of these two pathways on autophagy still needs to be further explored. Whenever plants are under conditions that are not conducive to their own growth, such as drought, salt stress, and virus invasion, autophagy will be activated, thereby improving plant resistance [121,122,123]. Under suboptimal growth conditions, autophagy is beneficial for improving crop yield [124], and the turnover of ATG proteins plays a decisive role in the level of autophagy. Therefore, the degradation mechanism of plant autophagy-related proteins needs to be studied more extensively in order that they can make a beneficial contribution to agricultural production.

## Figures and Tables

**Figure 1 ijms-23-07301-f001:**
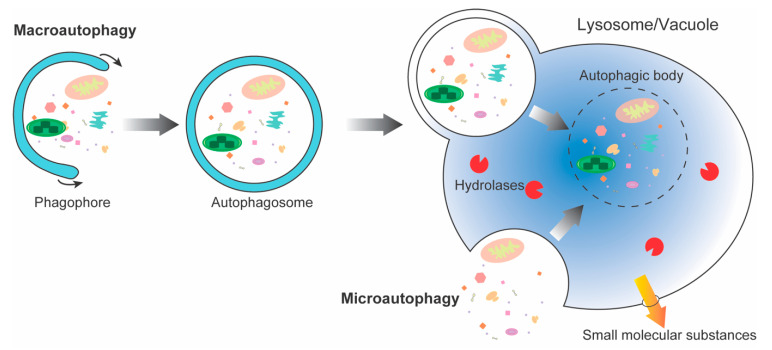
Morphological steps of microautophagy and macroautophagy in eukaryotes. Macroautophagy begins with the formation of a phagophore that encapsulates damaged organelles and discarded proteins. Then, through the extension of the vesicle forms a closed structure with a double membrane, which is called an autophagosome. Subsequently, the outer membrane of the autophagosome fuses with the lysosomal membrane (animals) or the tonoplast (yeast and plants) to release the autophagic body with only a single membrane. Finally, under the digestion of acid hydrolases, the cargoes were degraded into small molecular substances for recycling. Microautophagy is a process in which the lysosome/vacuole directly packages target substrates by membrane invagination to create the autophagic body.

**Figure 2 ijms-23-07301-f002:**
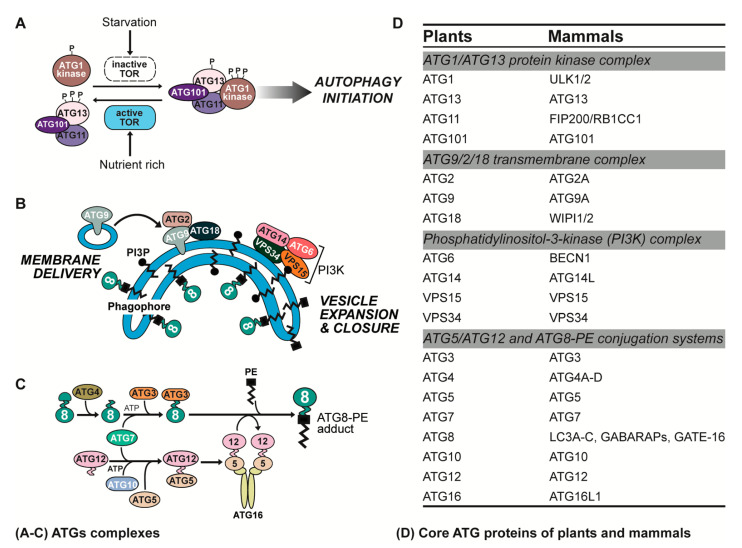
Core protein complexes of autophagy in plants and animals. (**A**) The ATG1/ATG13 protein kinase complex. When plants are under nutrient-rich conditions, TOR kinase hyperphosphorylates ATG13. While plants are placed in nutrient starved conditions, the TOR kinase is inactivated, resulting in the dephosphorylation of ATG13, which binds tightly to ATG1. Then, the ATG1 kinase activity is activated and autophosphorylation occurs to form the ATG1, ATG11, ATG13, and ATG101 complex, which results in upregulating autophagy. (**B**) The ATG9/2/18 transmembrane complex and PI3K complex. ATG9 delivers membrane source and mediates the extension of phagophore membrane. ATG2 and ATG18 play a synergistic role in this process. The PI3K protein complex promotes the nucleation of vesicles, which include ATG6, ATG14, VPS34, VPS15, and PI3P. (**C**) The ATG5-ATG12 and ATG8-PE ubiquitin-like conjugating systems. ATG12 was transferred to the target protein ATG5 with the help of ATG7 and ATG10. Subsequently, the ATG5-ATG12 complex combines with ATG16 to form an oligomeric complex, which participated in the esterification of ATG8. During the covalent binding stage of ATG8-PE, the cysteine protease ATG4 cleaves the C-terminus of ATG8. Subsequently, ATG8 is activated by ATG7 and transferred to ATG3 through a thioester bond. At last, with the help of ATG5-ATG12-ATG16 conjugate, ATG8 forms an ATG8-PE adduct with phosphatidylethanolamine. (**D**) Core ATG proteins of plants and mammals in four complexes.

**Figure 3 ijms-23-07301-f003:**
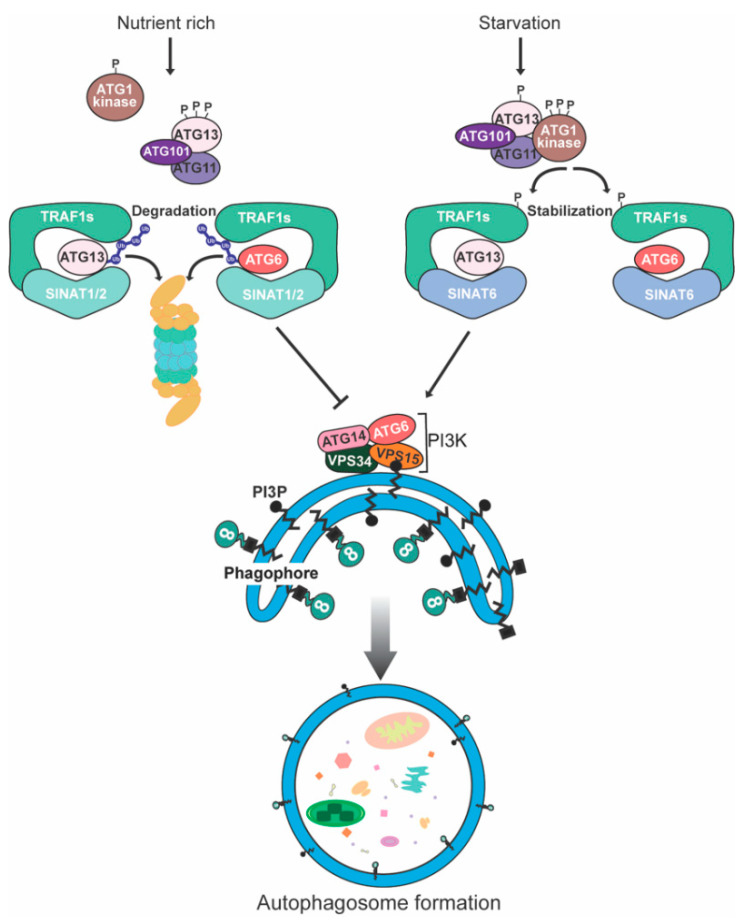
Degradation mechanisms of ATG6 and ATG13 in plants. During nutrient-rich conditions, the TRAF1s-SINAT1/SINAT2-ATG6 and TRAF1s-SINAT1/SINAT2-ATG13 TRAFasomes regulate the ubiquitination and proteasomal degradation of ATG6 and ATG13, which results in inhibiting autophagy. Under the condition of nutrient starvation, SINAT6 accumulates to form the TRAF1s-SINAT6-ATG6 and TRAF1s-SINAT6-ATG13 TRAFasomes, which maintain the stability of ATG6 and ATG13. Furthermore, ATG1 kinase phosphorylates TRAF1s to increase its stability.

**Table 1 ijms-23-07301-t001:** E3 ligases and ubiquitin chain types involved in ATGs degradation in yeast, mammalian, and plants.

Eukaryotes	ATGs	E3 Ligases	Ubiquitin Chain Types	References
Yeast	Atg9	Met30	Unknown	[49]
	Atg32	Rsp5	Unknown	[55]
Mammalian	ULK1	CUL3-KLHL20	K48	[56]
		BTRC	K48	[57]
		TRAF3	K48	[58]
		NEDD4L	K27 and K29	[59]
	ATG101	HUWE1	K48	[60]
	WIPI2	HUWE1	Unknown	[61]
	ATG14L	ZBTB16-CUL3-Roc1	Unknown	[62]
	BECN1	NEDD4	K11	[63]
		RNF216	K48	[64]
		CUL3-KLHL20	Unknown	[65]
		CUL3-KLHL38	K48	[65]
		TRAF6	K63	[66]
	VPS34	UBE3C	K29/K48 branched	[67]
		FBXL20-CUL1-SKP1	Unknown	[68]
	ATG16L1	Gigaxonin	K48	[69]
	LC3B	BIRC6	Single ubiquitin	[70]
	ATG3	PTK2	Unknown	[71]
Plants	ATG6	SINAT1/2	Unknown	[45]
	ATG13	SINAT1/2	K48	[72]
	SH3P2	XopL	Unknown	[73]

## Data Availability

Not applicable.

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
