# Peer review of "Degradation Mechanism of Autophagy-Related Proteins and Research Progress"

_ijms, 2022, doi:10.3390/ijms23137301_

Round 1
Reviewer 1 Report
In this review, Yanhui Zhou and collaborators gather the available information on the degradation of autophagy-related (ATG) proteins, which sets an additional layer of regulation for this catabolic process. The text is mainly focused on ubiquitin-mediated degradation, and it covers several eukaryotes, from yeasts to plants and animals.
This is for sure an interesting topic, and this review is a nice attempt to address it. However, I think there are some minor points the authors could improve before the manuscript should be accepted:
1) I think a simple “Degradative mechanism of autophagy-related proteins” would be a much more suitable title for this review, not only focusing on plants as not much is specifically discussed about research progress in plants.
In this regard, I would change the second section (“ATG complexes in plants”) so it also includes the other main eukaryotes addressed in the text (plants and animals).
2) Lines 33-34: Macroautophagy is a broad term that applies both to selective and non-selective autophagy. It only refers to the transportation of cytoplasmatic portions to the lysosomes by the double-membraned vesicles termed “autophagosomes”. Selective autophagy processes like “mitophagy” or “xenophagy” also involved the formation of autophagosomes, so they can be considered a subtype of macroautophagy.
3) Line 274: Proteins from the GABARAP subfamily (GABARAP, GATE-16 and ATG8L) can also be used as autophagosome markers. The LC3 subfamily just happens to be the one that is traditionally used.
4) The figures and the table are nice additions. However, I suggest improving the same table by including some information about E3 ligases and ubiquitination in yeasts and plants.
5) I also suggest talking about the degradation of yeast Atg proteins before discussing the same process in animal and plant cells (section 5 before sections 4 and 6).
6) Is there any information on pathogens using E3 ligases to evade autophagy when infecting animal cells?
7) The text is overall well-written, though I encourage a thoughtful revision of the use of the English language before it is finally published.
8) Some protein names are completely spelt with capital letters throughout the text and references (for example in line 225, SEQUESTOSOME). All of these should be corrected.
Author Response
Dear Editor,
We would like to thank you and the reviewers for the careful, constructive, and helpful reviews and the valuable comments and suggestions that improved the manuscript. We revised the manuscript accordingly, and detailed corrections are listed below point by point.
Please kindly note that in this responses file, the normal font = Reviewers’ comments; bold font = Authors’ Response. In the manuscript (MS) file, changes were made in tracking system.
Sincerely yours,
Dr. Hakim Manghwar
Reviewer#1
Comments and Suggestions for Authors
In this review, Yanhui Zhou and collaborators gather the available information on the degradation of autophagy-related (ATG) proteins, which sets an additional layer of regulation for this catabolic process. The text is mainly focused on ubiquitin-mediated degradation, and it covers several eukaryotes, from yeasts to plants and animals.
This is for sure an interesting topic, and this review is a nice attempt to address it. However, I think there are some minor points the authors could improve before the manuscript should be accepted:
1) I think a simple “Degradative mechanism of autophagy-related proteins” would be a much more suitable title for this review, not only focusing on plants as not much is specifically discussed about research progress in plants.
In this regard, I would change the second section (“ATG complexes in plants”) so it also includes the other main eukaryotes addressed in the text (plants and animals).
Response: Thank you so much for your suggestions. The title has been modified and the second section has been revised accordingly.
2) Lines 33-34: Macroautophagy is a broad term that applies both to selective and non-selective autophagy. It only refers to the transportation of cytoplasmatic portions to the lysosomes by the double-membraned vesicles termed “autophagosomes”. Selective autophagy processes like “mitophagy” or “xenophagy” also involved the formation of autophagosomes, so they can be considered a subtype of macroautophagy.
Response: Thank you so much. According to your suggestions, changes have been made.
3) Line 274: Proteins from the GABARAP subfamily (GABARAP, GATE-16 and ATG8L) can also be used as autophagosome markers. The LC3 subfamily just happens to be the one that is traditionally used.
Response: Thanks, it has been modified.
4) The figures and the table are nice additions. However, I suggest improving the same table by including some information about E3 ligases and ubiquitination in yeasts and plants.
Response: Thank you so much, it has been added.
5) I also suggest talking about the degradation of yeast Atg proteins before discussing the same process in animal and plant cells (section 5 before sections 4 and 6).
Response: Thank you so much for your advice, the order has been changed.
6) Is there any information on pathogens using E3 ligases to evade autophagy when infecting animal cells?
Response: Thank you so much. Circumsporozoite protein (CSP) is a surface protein of the Plasmodium sporozoite. Reference 96 introduced that cyclosporidium protein (CSP) downregulates LC3B through the UPS pathway, but the mechanism involved is unclear.
7) The text is overall well-written, though I encourage a thoughtful revision of the use of the English language before it is finally published.
Response: Thank you so much. Overall English language of whole manuscript has been improved.
8) Some protein names are completely spelt with capital letters throughout the text and references (for example in line 225, SEQUESTOSOME). All of these should be corrected.
Response: Thanks, all protein names have been corrected.
TRANSLATE with x EnglishArabic | Hebrew | Polish |
Bulgarian | Hindi | Portuguese |
Catalan | Hmong Daw | Romanian |
Chinese Simplified | Hungarian | Russian |
Chinese Traditional | Indonesian | Slovak |
Czech | Italian | Slovenian |
Danish | Japanese | Spanish |
Dutch | Klingon | Swedish |
English | Korean | Thai |
Estonian | Latvian | Turkish |
Finnish | Lithuanian | Ukrainian |
French | Malay | Urdu |
German | Maltese | Vietnamese |
Greek | Norwegian | Welsh |
Haitian Creole | Persian |
Reviewer 2 Report
Zhou Y et al.'s review titled "Degradation Mechanism of Autophagy-related Proteins and Research Progress in Plants" illustrated the recent progress in autophagy-related proteins that are degraded in yeast, animals, and plants. Generally, the review sounds interesting to the reader in the field but lacks the updated vital references. Please refer to those references I listed below and discuss them in the review accordingly. Also, the title is strange to me. Please think over the sentence structure. Suggest using "Degradation Mechanism of Autophagy-related Proteins in Plants."
· Yim WW, Mizushima N. Lysosome biology in autophagy. Cell discovery. 2020 Feb 11;6(1):1-2.
· Mizushima, Noboru, Eileen White, and David C. Rubinsztein. "Breakthroughs and bottlenecks in autophagy research." Trends in Molecular Medicine 27.9 (2021): 835-838.
· Mizushima N. The ATG conjugation systems in autophagy. Curr Opin Cell Biol. 2020 Apr;63:1-10. doi: 10.1016/j.ceb.2019.12.001. Epub 2019 Dec 31. PMID: 31901645.
· Zhang S, Hama Y, Mizushima N. The evolution of autophagy proteins - diversification in eukaryotes and potential ancestors in prokaryotes. J Cell Sci. 2021 Jul 1;134(13):jcs233742. doi: 10.1242/jcs.233742. Epub 2021 Jul 6. PMID: 34228793.
· Zhang S, Yazaki E, Sakamoto H, Yamamoto H, Mizushima N. Evolutionary diversification of the autophagy-related ubiquitin-like conjugation systems. Autophagy. 2022 Apr 15:1-16. doi: 10.1080/15548627.2022.2059168. Epub ahead of print. PMID: 35427200.
· Mizushima N. The ATG conjugation systems in autophagy. Curr Opin Cell Biol. 2020 Apr;63:1-10. doi: 10.1016/j.ceb.2019.12.001. Epub 2019 Dec 31. PMID: 31901645.
· SieÅ„ko K, Poormassalehgoo A, Yamada K, Goto-Yamada S. Microautophagy in Plants: Consideration of Its Molecular Mechanism. Cells. 2020 Apr 4;9(4):887. doi: 10.3390/cells9040887. PMID: 32260410; PMCID: PMC7226842.
· Bu F, Yang M, Guo X, Huang W, Chen L. Multiple Functions of ATG8 Family Proteins in Plant Autophagy. Front Cell Dev Biol. 2020 Jun 10;8:466. doi: 10.3389/fcell.2020.00466. PMID: 32596242; PMCID: PMC7301642.
· Pérez-Pérez ME, Lemaire SD, Crespo JL. The ATG4 protease integrates redox and stress signals to regulate autophagy. J Exp Bot. 2021 Apr 13;72(9):3340-3351. doi: 10.1093/jxb/erab063. PMID: 33587749.
· Yang C, Luo M, Zhuang X, Li F, Gao C. Transcriptional and Epigenetic Regulation of Autophagy in Plants. Trends Genet. 2020 Sep;36(9):676-688. doi: 10.1016/j.tig.2020.06.013. Epub 2020 Jul 13. PMID: 32674948.
· Lamark T, Johansen T. Mechanisms of Selective Autophagy. Annu Rev Cell Dev Biol. 2021 Oct 6;37:143-169. doi: 10.1146/annurev-cellbio-120219-035530. Epub 2021 Jun 21. PMID: 34152791.
· Zhang H. The Genetics of Autophagy in Multicellular Organisms. Annu Rev Genet. 2022 Jun 9. doi: 10.1146/annurev-genet-022422-095608. Epub ahead of print. PMID: 35679620.
· Nakatogawa H. Mechanisms governing autophagosome biogenesis. Nat Rev Mol Cell Biol. 2020 Aug;21(8):439-458. doi: 10.1038/s41580-020-0241-0. Epub 2020 May 5. PMID: 32372019.
Author Response
Dear Editor,
We would like to thank you and the reviewers for the careful, constructive, and helpful reviews and the valuable comments and suggestions that improved the manuscript. We revised the manuscript accordingly, and detailed corrections are listed below point by point.
Please kindly note that in this responses file, the normal font = Reviewers’ comments; bold font = Authors’ Response. In the manuscript (MS) file, changes were made in tracking system.
Sincerely yours,
Dr. Hakim Manghwar
Comments and Suggestions for Authors
Zhou Y et al.'s review titled "Degradation Mechanism of Autophagy-related Proteins and Research Progress in Plants" illustrated the recent progress in autophagy-related proteins that are degraded in yeast, animals, and plants. Generally, the review sounds interesting to the reader in the field but lacks the updated vital references. Please refer to those references I listed below and discuss them in the review accordingly. Also, the title is strange to me. Please think over the sentence structure. Suggest using "Degradation Mechanism of Autophagy-related Proteins in Plants."
- Yim WW, Mizushima N. Lysosome biology in autophagy. Cell discovery. 2020 Feb 11;6(1):1-2.
- Mizushima, Noboru, Eileen White, and David C. Rubinsztein. "Breakthroughs and bottlenecks in autophagy research." Trends in Molecular Medicine27.9 (2021): 835-838.
- Mizushima N. The ATG conjugation systems in autophagy. Curr Opin Cell Biol. 2020 Apr;63:1-10. doi: 10.1016/j.ceb.2019.12.001. Epub 2019 Dec 31. PMID: 31901645.
- Zhang S, Hama Y, Mizushima N. The evolution of autophagy proteins - diversification in eukaryotes and potential ancestors in prokaryotes. J Cell Sci. 2021 Jul 1;134(13):jcs233742. doi: 10.1242/jcs.233742. Epub 2021 Jul 6. PMID: 34228793.
- Zhang S, Yazaki E, Sakamoto H, Yamamoto H, Mizushima N. Evolutionary diversification of the autophagy-related ubiquitin-like conjugation systems. Autophagy. 2022 Apr 15:1-16. doi: 10.1080/15548627.2022.2059168. Epub ahead of print. PMID: 35427200.
- Mizushima N. The ATG conjugation systems in autophagy. Curr Opin Cell Biol. 2020 Apr;63:1-10. doi: 10.1016/j.ceb.2019.12.001. Epub 2019 Dec 31. PMID: 31901645.
- Sieńko K, Poormassalehgoo A, Yamada K, Goto-Yamada S. Microautophagy in Plants: Consideration of Its Molecular Mechanism. Cells. 2020 Apr 4;9(4):887. doi: 10.3390/cells9040887. PMID: 32260410; PMCID: PMC7226842.
- Bu F, Yang M, Guo X, Huang W, Chen L. Multiple Functions of ATG8 Family Proteins in Plant Autophagy. Front Cell Dev Biol. 2020 Jun 10;8:466. doi: 10.3389/fcell.2020.00466. PMID: 32596242; PMCID: PMC7301642.
- Pérez-Pérez ME, Lemaire SD, Crespo JL. The ATG4 protease integrates redox and stress signals to regulate autophagy. J Exp Bot. 2021 Apr 13;72(9):3340-3351. doi: 10.1093/jxb/erab063. PMID: 33587749.
- Yang C, Luo M, Zhuang X, Li F, Gao C. Transcriptional and Epigenetic Regulation of Autophagy in Plants. Trends Genet. 2020 Sep;36(9):676-688. doi: 10.1016/j.tig.2020.06.013. Epub 2020 Jul 13. PMID: 32674948.
- Lamark T, Johansen T. Mechanisms of Selective Autophagy. Annu Rev Cell Dev Biol. 2021 Oct 6;37:143-169. doi: 10.1146/annurev-cellbio-120219-035530. Epub 2021 Jun 21. PMID: 34152791.
- Zhang H. The Genetics of Autophagy in Multicellular Organisms. Annu Rev Genet. 2022 Jun 9. doi: 10.1146/annurev-genet-022422-095608. Epub ahead of print. PMID: 35679620.
- Nakatogawa H. Mechanisms governing autophagosome biogenesis. Nat Rev Mol Cell Biol. 2020 Aug;21(8):439-458. doi: 10.1038/s41580-020-0241-0. Epub 2020 May 5. PMID: 32372019.
Response: Thank you so much for your advice. The title has been modified. In addition, the updated references you listed have been added in the manuscript.
TRANSLATE with x EnglishArabic | Hebrew | Polish |
Bulgarian | Hindi | Portuguese |
Catalan | Hmong Daw | Romanian |
Chinese Simplified | Hungarian | Russian |
Chinese Traditional | Indonesian | Slovak |
Czech | Italian | Slovenian |
Danish | Japanese | Spanish |
Dutch | Klingon | Swedish |
English | Korean | Thai |
Estonian | Latvian | Turkish |
Finnish | Lithuanian | Ukrainian |
French | Malay | Urdu |
German | Maltese | Vietnamese |
Greek | Norwegian | Welsh |
Haitian Creole | Persian |
Round 2
Reviewer 2 Report
The authors addressed my comments, and I have no further questions regarding this manuscript.